# Dyes Adsorption Behavior of Fe_3_O_4_ Nanoparticles Functionalized Polyoxometalate Hybrid

**DOI:** 10.3390/molecules24173128

**Published:** 2019-08-28

**Authors:** Jie Li, Chen Si, Haiyan Zhao, Qingxi Meng, Bowen Chang, Mingxue Li, Hongling Liu

**Affiliations:** Henan Key Laboratory of Polyoxometalates Chemistry, Institute of Molecular and Crystal Engineering, College of Chemistry and Chemical Engineering, Henan University, Kaifeng 475004, Henan, China

**Keywords:** polyoxometalate, Fe_3_O_4_, nanocomposite, adsorption, dyes

## Abstract

The magnetic adsorbent, Fe_3_O_4_@[Ni(HL)_2_]_2_H_2_[P_2_Mo_5_O_23_]·2H_2_O (Fe_3_O_4_@**1**), is synthesized by employing the nanoparticles Fe_3_O_4_ and polyoxometalate hybrid **1**. Zero-field-cooled (ZFC) and field-cooled (FC) curves show that the blocking temperature of Fe_3_O_4_@**1** was at 120 K. Studies of Fe_3_O_4_@**1** removing cationic and anionic dyes from water have been explored. The characterization of Fe_3_O_4_@**1**, effects of critical factors such as dosage, the concentration of methylene blue (MB), pH, adsorption kinetics, isotherm, the removal selectivity of substrate and the reusability of Fe_3_O_4_@**1** were assessed. The magnetic adsorbent displayed an outstanding removal activity for the cationic dye at a broad range of pH. The adsorption kinetics and isotherm models revealed that the adsorption process of Fe_3_O_4_@**1** was mainly governed via chemisorption. The maximum capacity of Fe_3_O_4_@**1** adsorbing substance was 41.91 mg g^−1^. Furthermore, Fe_3_O_4_@**1** showed its high stability by remaining for seven runs of the adsorption-desorption process with an effective MB removal rate, and could also be developed as a valuable adsorbent for dyes elimination from aqueous system.

## 1. Introduction

With the development of world’s population and socioeconomics, fresh and clean water has become one of the major concerns of 21^st^ century. The pollutants in aquatic ecosystems have caused serious environmental problems and threatened public health. Developing sustainable and economical technologies for treating wastewater process faces true challenges [1,2,3]. The pollution caused by organic dyes has been a great menace to the water ecosystem and human health [4,5]. Many researches have been implemented for the organic dyes removal, such as electrochemical oxidation, coagulation, adsorption, membrane filtration, biological schemes and photocatalytic degradation [6,7,8,9]. Among these strategies, adsorption is recognized as a promising approach due to its operating environment, simple procedures and high efficiency [10]. Many adsorbents have been used for the removal of dyes from aqueous solutions such as activated carbon, chitosan particles, aerogels, etc. [11,12,13]. However, it should be noted that the traditional adsorption materials have low removal efficiency and limited recycling ability. Therefore, developing effective adsorbents to remove the pollutants from wastewater is of great concern to researchers.

Phosphomolybdates are key to this issue due to their special structure and potential application in water pollution treatment [14,15,16]. One of the most prominent structures of phosphomolybdate is [P_2_Mo_5_O_23_]^6−^. Transition metal ions also have possible application in the dyes removal by coordinating with dyes containing –N=N–, –C=C– and heterocyclic compounds [17,18]. In particular, nickel is a vital element to contribute to the removal of dyes [19,20]. Additionally, thiosemicarbazones are also beneficial for the dyes removal by forming hydrogen bonds and π–π stacking interactions [21]. Thus, the combination of [P_2_Mo_5_O_23_]^6−^, nickel ion and 2-acetylpyrazine-thiosemicarbazone at a molecular will contribute to the dye removal by synergic effect. However, it is difficult to separate the hybrid phase from the liquid phase. Moreover, the non-recyclability may limit the application of the hybrid on a large scale. Recently, magnetic separation has gained increasing attention due to the fast and noncontact magnetic response by external magnetic field [22,23,24]. Fe_3_O_4_ nanoparticles are ferrite magnetic materials, which have strong magnetic properties. They have been taken as a carrier to synthesize nanocomposites, which can be easily removed and regained under a magnetic field [25]. 

Taking the above considerations, a Fe_3_O_4_-based magnetic adsorbent Fe_3_O_4_@**1** was synthesized and characterized. The adsorption activity was investigated by adsorbing MB, rhodamine B (RhB) and methyl orange (MO) dissolving in water. Fe_3_O_4_@**1** has demonstrated an effective selectivity for adsorbing cationic dyes. Moreover, the effects of some key factors such as the adsorbent dosage, the concentration of MB and pH on adsorption performance were studied. The adsorption behavior of Fe_3_O_4_@**1** was explored by adsorption kinetics and isotherm models, along with its recyclability and stability for multiple cycles of adsorption-desorption process.

## 2. Results and Discussion

### 2.1. Structural Descriptions

**1** comprises a strandberg-type [P_2_Mo_5_O_23_]^6−^ anion, two [Ni(HL)_2_]^2+^, two protons and two crystal water molecules (Figure 1a). [P_2_Mo_5_O_23_]^6−^ and [Ni(HL)_2_]^2+^ are bound together by hydrogen bonds, as shown by dotted orange lines in Figure 1a and electrostatic interaction. The anion [P_2_Mo_5_O_23_]^6−^ possesses two central PO_4_ tetrahedrons which combine the edge-sharing MoO_6_ octahedrons into a single cluster. In cationic [Ni(HL)_2_]^2+^, two Ni^2+^ show the similar coordination geometry by coordinating four N atoms and two S atoms from the two ligands HL, which shows the same coordination patterns of distorted octahedral geometries. The amino and pyrazine groups in ligand HL will conduce to form the hydrogen bonds and π–π stacking interactions with dyes. Figure 1b shows the three-dimensional network of **1**. The anionic [P_2_Mo_5_O_23_]^6−^ and amino-included ligands coexisting in a molecule are conducive for the interaction with cationic dyes by electrostatic interaction and hydrogen bonds.

### 2.2. FTIR Spectra

In the spectrum of Fe_3_O_4_@**1**, the characteristic peaks of **1** and Fe_3_O_4_ are included (Figure 2). Both Fe_3_O_4_@**1** and Fe_3_O_4_ exhibit the same peak at 588 cm^−1^, matching with the Fe–O band. The broad bands around 3492 cm^−1^ are attached to the O–H stretching vibration from water molecules [26]. Two characteristic peaks at 3280 and 3154 cm^−1^ are assigned to N–H stretching vibrations. The peak at 1614 cm^−1^ is associated with *ν*(C=N) bonds. The peak at 1219 cm^−1^ corresponds the *ν*(P–O) bands. The peaks between 950 and 853 cm^−1^ are appointed to the terminal Mo=O vibration. The range of 859–664 cm^−1^ is attributed to Mo–O–Mo vibration [27]. Therefore, **1** and Fe_3_O_4_ are included in Fe_3_O_4_@**1**.

### 2.3. UV-Vis Spectra

The spectra of **1** and Fe_3_O_4_@**1** show four absorption peaks in water (Figure 3). The peaks at 212 and 233 nm correspond to the charge-transfer of O_t_→Mo, and the peaks at 312 and 410 nm are assigned to the charge-transfer of O_b_→Mo [28]. There are no absorption peaks obtained for Fe_3_O_4_ nanoparticles. These results show that **1** exists in Fe_3_O_4_@**1**. 

### 2.4. XPS Characterization

The XPS analysis was recorded to further identify the successful preparation of Fe_3_O_4_@**1**. Results indicate that the presence of C 1s, N 1s, S 2p, P 2p, Mo 3d, Fe 2p and Ni 2p are at the sample surface. The corresponding detailed peaks of Fe and Mo are shown in Figure 4. The binding energy (BE) was evaluated by C 1s peak (284.8 eV). The BE of Fe 2p_1/2_ and Fe 2p_3/2_ are around 725.0 and 711.1 eV, respectively. The BE at 708.8 and 710.3 eV conforms to Fe^2+^ 2p_3/2_ and Fe^3+^ 2p_3/2_, respectively (Figure 4a) [29,30]. The XPS spectrum of Fe_3_O_4_@1 for Mo atoms gave two peaks with the BE of 235.2 and 232.2 eV respectively referring to Mo 3d_3/2_ and Mo 3d_5/2_ (Figure 4b) [31]. It indicated that Fe_3_O_4_ and **1** did coexist in Fe_3_O_4_@**1**. 

### 2.5. Magnetic Properties of Fe_3_O_4_@**1**

Figure 5a,b demonstrates the hysteresis curves of Fe_3_O_4_@**1**. Under the 10,000 Oe magnetic field, Fe_3_O_4_@**1** exhibits ~35 Oe coercivity and ~43 emu g^−1^ magnetization at 300 K, while the coercivity of ~280 Oe and the magnetization of ~47 emu g^−1^ are at 5 K. The hysteresis curves of Fe_3_O_4_@**1** indicates that Fe_3_O_4_@**1** change from soft ferromagnetic at 300 K to ferromagnetic at 5 K. Meanwhile, the magnetization increases when the temperature decreases. Under 500 Oe, the curves of ZFC and FC between 5 and 300 K were also studied (Figure 5c). It turned out that the blocking temperature of Fe_3_O_4_@**1** was circa 120 K [32]. Under the temperature, the FC curve is almost flat, whereas the ZFC curve falls sharply. Figure 5d represents the segregation and diffusion process of Fe_3_O_4_@**1** in water. The solution changed from brown uniform diffusion to achromous transparence under the influence of a magnetic field. After removing the magnetic field, the collected Fe_3_O_4_@**1** could be easily dispersed with agitation. The segregation and diffusion process could be repeated for many times, which further indicated that the Fe_3_O_4_@**1** were magnetic. In addition, the process also indicated that hybrids **1** and Fe_3_O_4_ were successfully combined together as **1** was nonmagnetic. It should be noted that the magnetic responsiveness and redispersibility of Fe_3_O_4_@**1** play important roles in the repeatability of adsorption application.

### 2.6. Morphology and Particle Size Analyses

Figure 6a represents the TEM image of pure Fe_3_O_4_@**1**. It is clear that pure Fe_3_O_4_@**1** has uniform particle morphology of roundness shape with an average size of 20.8 nm. The distribution was properly illustrated by the Gaussian function (Figure 6b). Figure 6c represents the HRTEM images of Fe_3_O_4_@**1** alone, with the 2.53 Å spacing matching the (311) reflection of Fe_3_O_4_ nanoparticle. Even though **1** covered on the Fe_3_O_4_ exterior was hardly classified, the coarseness and expanded thickness of the periphery indicated that **1** has been profitably incorporated on Fe_3_O_4_. Furthermore, elemental mappings (Figure 6d–l) illustrate the distribution of the elements C, O, Ni, Mo, N, Fe, P and S in the Fe_3_O_4_@**1** sample, indicating that **1** and Fe_3_O_4_ coexist in Fe_3_O_4_@**1**. These further confirm the successful formation of Fe_3_O_4_@**1** nanocomposites. 

### 2.7. The XRD Patterns and BET Analysis

The pattern of Fe_3_O_4_@**1** indicates that the peaks of **1** and Fe_3_O_4_ are included (Figure 7a). The diffraction peaks of Fe_3_O_4_ correspond to the stand card of JCPDS No. 75-0449. The average particle size of Fe_3_O_4_@**1** is 19.7 nm. The value is calculated by the Debye-Scherrer equation. The numerical value is approximate to the TEM analysis. The result of PXRD analysis demonstrates that **1** and Fe_3_O_4_ are well combined. 

Figure 7b exhibits the N_2_ adsorption-desorption and the aperture distribution curve of Fe_3_O_4_@**1**. According to the classification of IUPAC, Fe_3_O_4_@**1** shows a symbolic type IV isotherm curve with H1 hysteresis loop [33]. Besides, the adsorption curve is closer but leveled below the desorption curve. Therefore, the Fe_3_O_4_@**1** is typical mesoporous structure [34]. The generated pores are conducive to the removal of MB [35]. For the Fe_3_O_4_@**1**, the BET surface area was 15.05 m^2^ g^−1^, the pore volume was 0.07 cm^3^ g^−1^ and the average diameter of BJH pore was 3.83 nm, respectively.

### 2.8. Dye Adsorption Experiments

In recent decades, nanocomposites have attracted increasing attention in adsorbing organic dyes from wastewater. To determine whether Fe_3_O_4_@**1** is a suitable adsorbent for organic dyes, MB was used as an exemplary role because of its extensive usage in academic studies and numerous industries. Fe_3_O_4_@**1** was put into 10 mL water of MB (15 mg L^−1^) at pH 6.98 for 3 h. Fe_3_O_4_@**1** displayed a removal rate of 97.84% and adsorption capacity of circa 41.91 mg g^−1^ on MB. After adsorption, the MB solution almost turned colorless. Some major aspects which might influence the adsorption activity, such as the adsorbent dosage, MB concentration, pH, adsorption kinetics, isotherm models and substrate selectivity were studied.

The optimum usage amount of Fe_3_O_4_@**1** was studied by injecting multifarious amounts (2, 4, 6, 8, 10, 12 and 14 mg) to aqueous MB of 15 mg L^−1^ (experimental condition: pH, 7; temperature, 25 °C; volume, 10 mL; time, 3 h). As shown in Figure 8a, the removal rate reached 97.84% when 12 mg of the adsorbent was used. Thus, 12 mg/10 mL was concluded to be the optimum usage/volume proportion for the adsorption process, which was applied for the following experiments. The influence of MB concentration was explored as well (Figure 8b). While increasing MB concentration from 5 to 30 mg L^−1^ (5, 10, 15, 20, 25 and 30 mg L^−1^), the removal rate changed to 95.02%, 97.53%, 97.84%, 97.76%, 95.81% and 94.70%, accordingly. Under low concentrations, saturation was achieved after 30 minutes. Indeed, MB was almost completely removed within 3 h with different concentrations. Therefore, 15 mg L^−1^ is considered to be the optimum concentration for the adsorption process.

The influence of solution pH was investigated as it was a crucial aspect that dictated the adsorption property due to its relationship with the extent of ionization with substrate and apparent charge of adsorbent. As demonstrated in Figure 9, Fe_3_O_4_@**1** represents varying degrees of affinity towards MB at the range of 3–13. The MB adsorption efficiency retained a relatively high situation, while the initial values were between 5 and 11. The activity of Fe_3_O_4_@**1** at the initial pH value of 7 was the best. At a high pH value of 13, the removal rate was only 36.57%, which might be the reason that the structure of **1** of Fe_3_O_4_@**1** was damaged by the excess OH^−^. Therefore, further studies were explored under neutral pH condition (pH circa 7.0). 

The adsorption kinetics was analyzed using the pseudo-first-order and pseudo-second-order model. Figure 10a,b shows the correlation coefficient (*R^2^*) of the above two models, respectively. It can be observed that the pseudo-second-order model showed a larger *R^2^* than that of the pseudo-first order, demonstrating that the adsorption process was mainly governed via chemisorption [36].

The adsorption isotherms of MB at 298 K were investigated with the adsorbents of 12 mg for 3 h. Langmuir and Freundlich isotherm models were utilized to further study the adsorption capacities of adsorbents and the interaction between adsorbate and adsorbent. The two isotherms models are described in Equations S5 and S6, respectively [37]. As shown in Figure 10c,d, it can be found that the Langmuir isotherm model was more reasonable for the description of the adsorption behavior of the Fe_3_O_4_@**1** toward MB, which indicated that the recognition tendency was mainly because of the charge interactions referring to negative charge of **1** and the cationic of MB, hydrogen bonds and π–π stacking interactions [10,38,39]. The comparison experiments of Fe_3_O_4_, **1** and Fe_3_O_4_@**1** also revealed that **1** of Fe_3_O_4_@**1** might be the active site in adsorption process. The Fe_3_O_4_ nanoparticles of Fe_3_O_4_@**1** were beneficial for the recovery of adsorbents (Appendix A). In order to affirm this assumption, cationic dye rhodamine B (RhB) and anion dye methyl orange (MO) were adopted to examine the adsorption behavior of Fe_3_O_4_@1 at the same condition. It turned out that the removal rate of RhB was 91.22%, while Fe_3_O_4_@**1** has little effect on MO removal (Figure 10e,f). These results can be regarded as a solid support for the removal of cationic dyes in the presence of Fe_3_O_4_@**1**.

The reusability of nanocomposites plays a crucial role in developing reliable, economic and sustainable applications [40,41,42]. The reusability of Fe_3_O_4_@**1** was investigated by means of retired adsorption cycles on the same sample (MB solution, 15 mg L^−1^). In the process, the adsorbent was isolated by a magnet after every adsorption reaction and was washed by ethanol for six times and dried at 60 °C for 2 h. The adsorbent almost preserved the same adsorption capacity, even after seven runs with a small decline in the yield to 97.84%, 96%, 95.4%, 94.75%, 93.36%, 91.34% and 90.44%, respectively (Figure 11a). The variation of adsorption capacities between the first and seventh adsorption cycle was found to be less than 8%. The slight decrease might be caused by the loss of the used adsorbents in recovery process. It should be mentioned that further optimization of the elution might improve the recycling. Interestingly, the FTIR and XPS spectra were nearly identical before and after seven cycles of adsorption reaction, which demonstrated the high stability of Fe_3_O_4_@**1** (Figure 11b,c). Thus, it is concluded that Fe_3_O_4_@**1** has a good recyclability and remarkable stability in the removal of MB at present experiments. This aspect is very important in practical applications of economic and stringent ecological demands for sustainability. 

## 3. Experiment

### 3.1. Materials and Methods

All reagents and chemicals were acquired from commercial sources. Nickel(II) perchlorate hexahydrate (Ni(ClO_4_)_2_·6H_2_O, reagent grade), sodium molybdate dihydrate (Na_2_MoO_4_*·*2H_2_O, 99%), phosphoric acid (H_3_PO_4_, 85%), 1,2-hexadecanediol (C_14_H_29_CH(OH)CH_2_(OH), 98%), dioctyl ether (C_8_H_17_OC_8_H_17_, 96%), iron(III) acetylacetonate (Fe(acac)_3_, 98%), Poly(ethylene glycol)-block-poly(propylene glycol)-block-poly(ethyleneglycol) (PEO-PPO-PEO, Mr = 5800) were purchased from J&K Scientific Ltd. (Beijing, China). Methanol (CH_3_OH, 99.5%), ethanol (CH_3_CH_2_OH, 99.7%), 2-acetylpyrazine (C_4_N_2_H_3_COCH_3_, 99%) and thiosemicarbazide (CH_5_N_3_S, 99%) were purchased from Aldrich (Shanghai, China). The Fe_3_O_4_ nanoparticles were prepared according to the article scheme [43]. X-ray powder diffraction patterns (PXRD) were tested with a Cu Kα radiation. A Flash 2000 analyzer was employed to test C, H, and N. Ni and Mo were confirmed by an inductively-coupled plasma spectrometer (Leaman, city, state abbrev if USA, Country). A Nicolet FTIR 360 spectrometer (company, city, state abbrev if USA, Country) was used to obtain the infrared (FTIR) spectrum. Ultraviolet-visible (UV-Vis) spectrum was captured by a TU–1900 spectrometer (company, city, state abbrev if USA, Country). The physical property measurement system (PPMS) and vibrating sample magnetometer (VSM, company, city, state abbrev if USA, Country) were used to explore the magnetic property. Transmission electron microscopy (TEM, JEOL2010F) and high resolution (HRTEM) were collected on a JEOL2010F (company, city, state abbrev if USA, Country). X-ray photoelectron spectrum (XPS) was studied by Al *Kα* X-ray as the excitation resource by a Thermo ESCALAB 250XI photoelectron spectrometer (company, city, state abbrev if USA, Country). 

### 3.2. Preparations of [Ni(HL)_2_]_2_H_2_[P_2_Mo_5_O_23_]·2H_2_O *(**1**)*

A mixture of Ni(ClO_4_)_2_·6H_2_O (0.091 g, 0.25 mmol), 2-acetylpyazine-thiosemicarbazone (0.096 g, 0.5 mmol), methanol and water solution (25 mL, 3:2) was stirred for 30 minutes at 60 °C. After cooling, the above solution was mixed with 10 mL water with Na_2_MoO_4_·2H_2_O (0.24 g, 1.0 mmol). The pH of the solution was maintained around 3.0 with concentrated H_3_PO_4_. Such obtained solution was stirred at 60 °C for another 30 minutes, then cooled and filtrated. The filtrate evaporated slowly at room temperature in an open beaker and brown rod crystals were obtained after five days. Yield: circa 65.80% (based on Ni). Elemental analysis for C_28_H_42_Mo_5_N_20_Ni_2_O_25_P_2_S_4_. Calculation (%): C 18.22, H 2.29, N 15.17, Ni 6.36, Mo 25.99; Found (%): C 18.36, H 2.60, N 15.41, Ni 6.45, Mo 26.14. 

### 3.3. Preparation of Nanocomposites Fe_3_O_4_@**1**

Fe_3_O_4_ (7.5 mg) and **1** (50 mg) were added to a 10 mL solution (V_water_:V_ethanol_ = 1:1) which was then dispersed with ultrasonic for 10 h. During the ultrasound process, the solution is thermostatic (30 °C). Then, the materials (Fe_3_O_4_@**1**) were segregated by a magnet and washed with water. The weight ratio of 1 in Fe_3_O_4_@1 is 89.57% (Appendix A).

### 3.4. X-Ray Crystallographic Study

Data of **1** were carried out on a Bruker D8 SMART APEX-II CCD X-ray diffraction with graphite-monochromated Mo K*α* radiation (λ = 0.71073 Å) (company, city, state abbrev if USA, Country). Lorentz and polarization rectification were utilized, and an applied multi-scan assimilation rectification was achieved with the SADABS scheme. The structure was determined by direct methods with the SHELXTL program package [44,45]. All atoms were anisotropically refined—except for hydrogen. Hydrogen atoms were fixed geometrically. The detailed crystallographic data are given in Appendix A. The CCDC number of **1** is 1918602.

### 3.5. Adsorption Experiments

The adsorption experiments were conducted in the dark at room temperature. Typically, 12 mg of the adsorbents were added to 10 mL MB solution (15 mg L^−1^). The reaction suspension (4 mL) was taken out from the beaker periodically by a pipette. The isolated clear solution was explored by a UV-vis spectrophotometer. The removal efficiency and adsorption capacity (*q*_e_) of adsorbents was calculated by Equations S1 and S2 [46]. The experimental info was plotted using pseudo-first order and pseudo-second order kinetic model (Equations S3 and S4) [47].

## 4. Conclusions

In conclusion, a magnetic adsorbent Fe_3_O_4_@**1** was successfully prepared and systematically characterized. The adsorption behavior of Fe_3_O_4_@**1** for organic dyes from aqueous solution was explored with MB as a model substrate. The ZFC-FC measurements showed that the blocking temperature of Fe_3_O_4_@**1** was at 190 K. Although the low BET surface area is 15.05 m^2^ g^−1^, Fe_3_O_4_@**1** displayed an effective adsorption efficiency of 97.84% for MB. The adsorption of MB depends on adsorbent usage, initial MB concentration and solution pH. Moreover, the potential reason of adsorption was discussed as well according to results from adsorption experiments, which revealed that the adsorption of MB through Fe_3_O_4_@**1** was mainly caused by chemisorption. The maximal adsorption capacity of Fe_3_O_4_@**1** was 41.91 mg g^−1^. Furthermore, Fe_3_O_4_@**1** demonstrated a great performance for selectively adsorbing cationic dyes. Lastly, the adsorbent can last at least for seven cycles of adsorption-desorption process with high stability, indicating that Fe_3_O_4_@**1** has potential application in cationic dyes removal. 

## Figures and Tables

**Figure 1 molecules-24-03128-f001:**
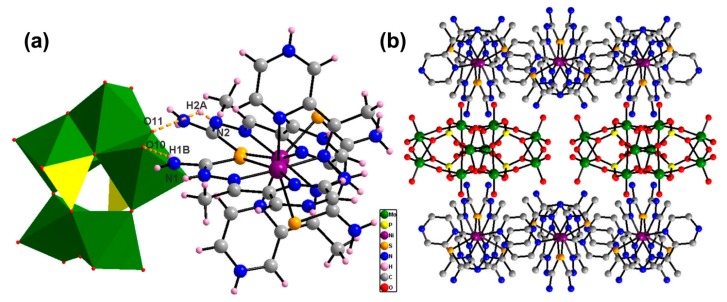
(**a**) Polyhedral/ ball-stick representations and (**b**) ball-stick representation of 3D network of **1**.

**Figure 2 molecules-24-03128-f002:**
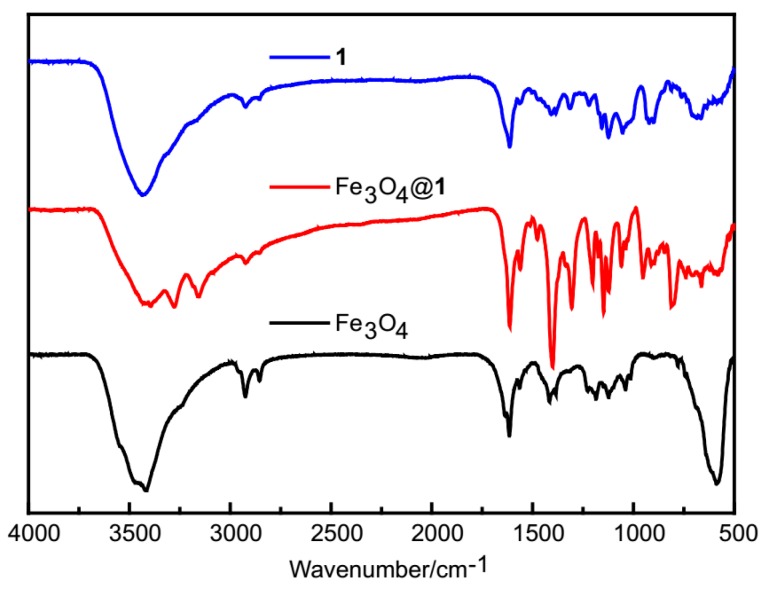
The FTIR spectra of **1**, Fe_3_O_4_@**1** and Fe_3_O_4_.

**Figure 3 molecules-24-03128-f003:**
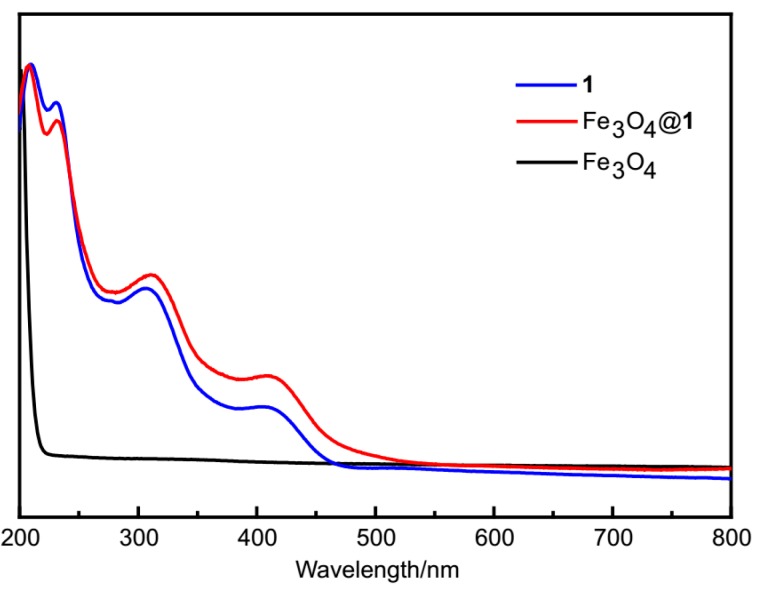
The UV-Vis spectra of **1**, Fe_3_O_4_@**1** and Fe_3_O_4_.

**Figure 4 molecules-24-03128-f004:**
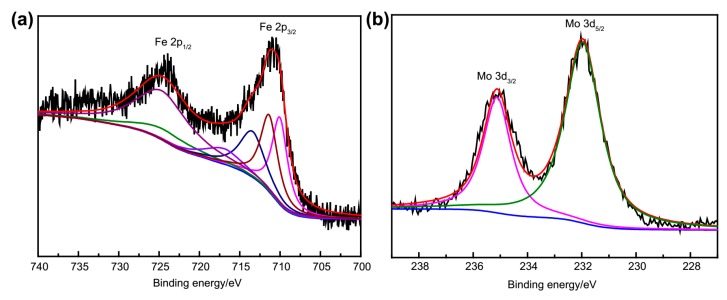
XPS spectra of (**a**) Fe 2p and (**b**) Mo 3d of Fe_3_O_4_@**1**.

**Figure 5 molecules-24-03128-f005:**
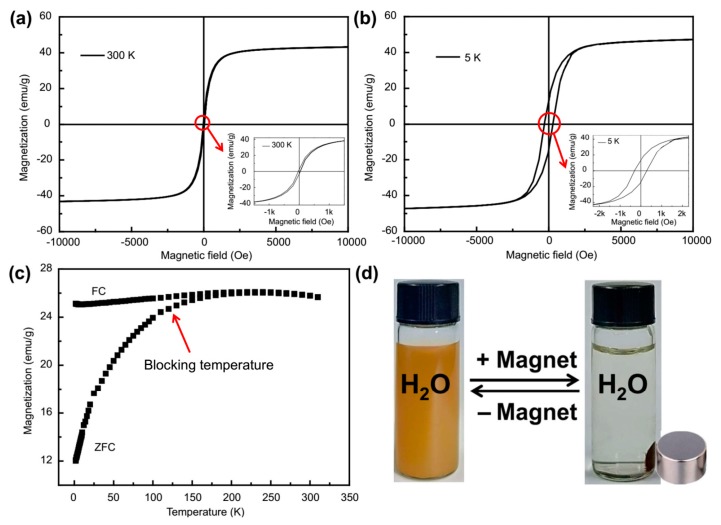
Hysteresis curves at (**a**) 300 K and (**b**) 5 K, (**c**) ZFC and FC curves and (**d**) the segregation and diffusion process of Fe_3_O_4_@**1**.

**Figure 6 molecules-24-03128-f006:**
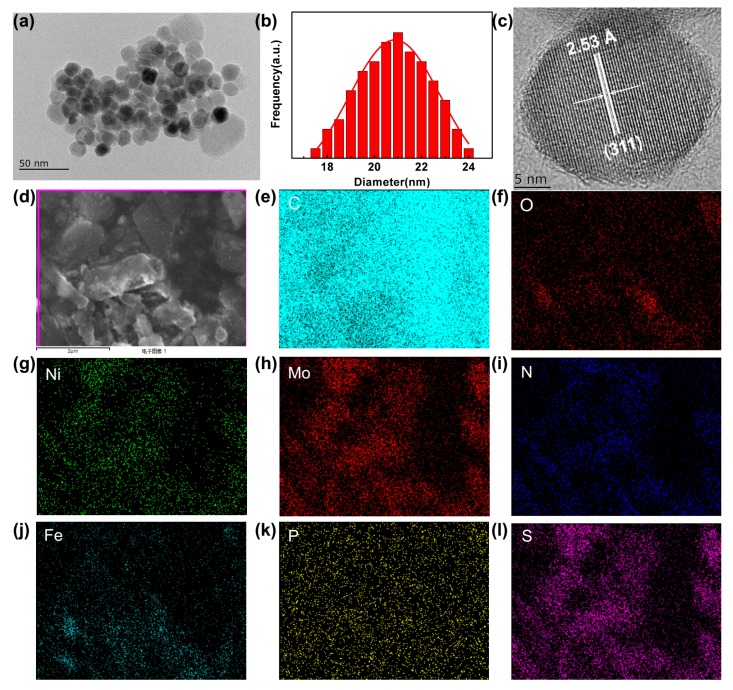
(**a**) TEM image, (**b**) particle size histogram, (**c**) HRTEM image, (**d**) STEM image and (**e**–**l**) corresponding elemental mappings of Fe_3_O_4_@**1**.

**Figure 7 molecules-24-03128-f007:**
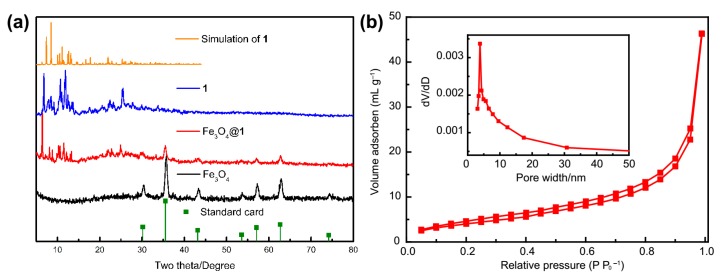
(**a**) The XRD diffraction analyses, (**b**) N_2_ adsorption-desorption curve. Inset figure: aperture distribution curve of Fe_3_O_4_@**1**.

**Figure 8 molecules-24-03128-f008:**
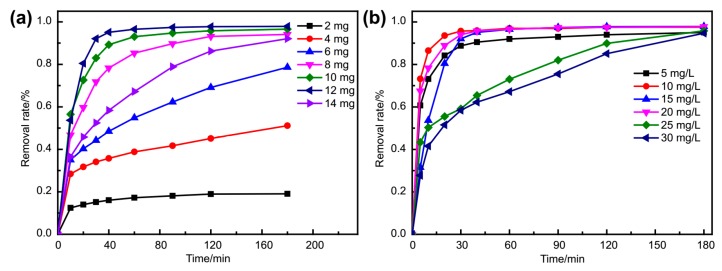
The influence of (**a**) adsorbent usage and (**b**) MB concentration.

**Figure 9 molecules-24-03128-f009:**
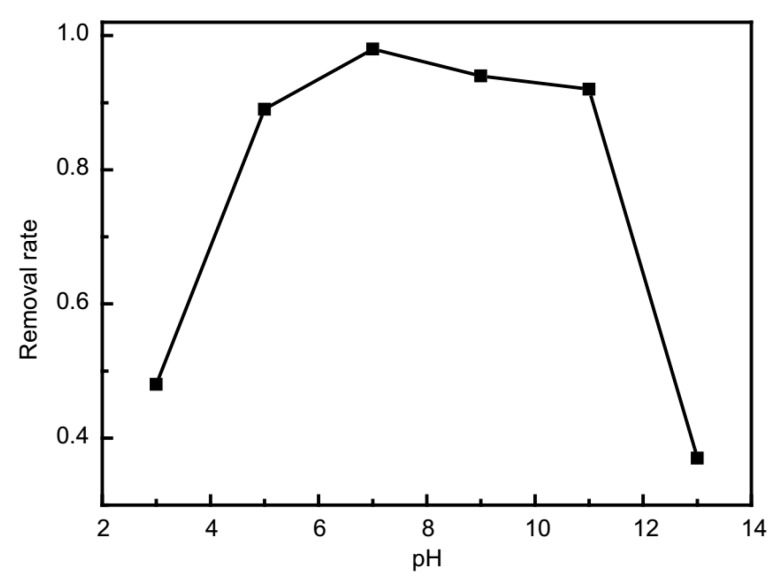
The influence of pH on MB adsorption over Fe_3_O_4_@**1** adsorbent.

**Figure 10 molecules-24-03128-f010:**
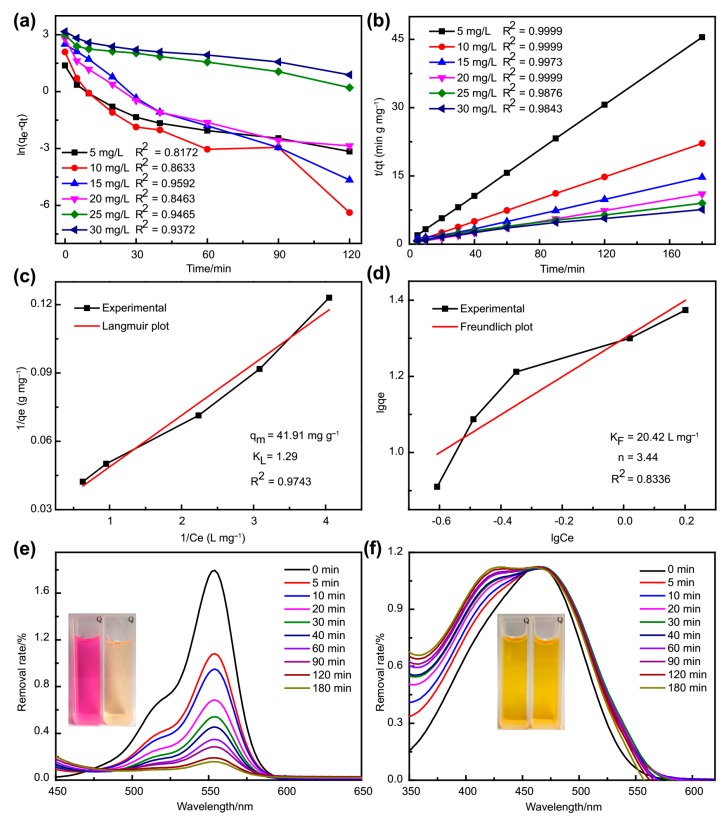
Plots for (**a**) pseudo-first-order, (**b**) pseudo-second-order, (**c**) Langmuir and (**d**) Freundlich isotherm model for the adsorption of MB onto Fe_3_O_4_@**1**. The adsorption spectra of (**e**) RhB and (**f**) MO by Fe_3_O_4_@**1**.

**Figure 11 molecules-24-03128-f011:**
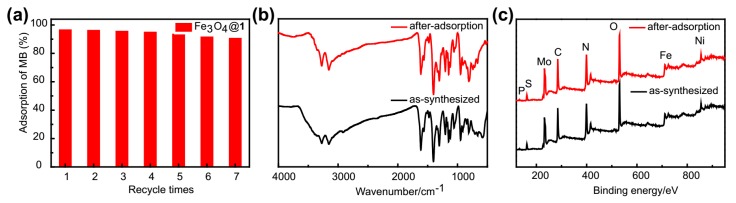
(**a**) The reusability studies of Fe_3_O_4_@**1**. The after-adsorption and as-synthesized (**b**) FTIR and (**c**) XPS spectra of Fe_3_O_4_@**1**.

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
