# Peer review of "Dyes Adsorption Behavior of Fe_3_O_4_ Nanoparticles Functionalized Polyoxometalate Hybrid"

_molecules, 2019, doi:10.3390/molecules24173128_

Round 1
Reviewer 1 Report
The manuscript deals with the development of a magnetic nano-composite and its characterization with further application for dyes removal. The work should state its novelty and scientific significance. Following corrections are required:
The language of the manuscript is poor and requires major revision before possible publication The Introduction should explicitly state about why the authors require the necessity to develop a nano-composite from phosphomolybdate. Also, stating about previous research work done on phosphomolybdate as an adsorbent. more references are required in Introduction The following references need to be incorporated:water Research 145, 741-756 (2019), Journal of colloid and interface science 493, 228-243 (2017), Journal of environmental chemical engineering 4, 4342-4353 (2016) How have the authors carried out structural determination as explained in Section 3.1 More explanation required on FTIR and UV/Vis discussion STEM images are not clear How did the authors determine BET surface area, pore volume and diameter. Authors need to comment on mechanism of dye adsorption based on the FTIR, pH results and XPS study results "Comparision experiments...." in page 9 need to be explained How have the authors concluded that chemisorption is the mechanism of dye adsorption. More explanation is required before concluding
Author Response
Response: Thank you for your letter and comments concerning our manuscript. These comments are all valuable and very helpful for revising and improving our paper. Revised portion are marked in red in the paper. First, we feel sorry for our poor writings, however, we do invite an English speaker to polish our article. We have done our best to improve the quality of English and typing errors. And we hope the revised manuscript could be acceptable for you. As we described in the introduction, phosphomolybdates are of great concerns due to their special structure and potential application in water pollution treatment, which is the reason for developing a nanocomposite from phosphomolybdate. The related references water Research 145, 741–756 (2018), Journal of colloid and interface science 493, 228–243 (2017) and Journal of environmental chemical engineering 4, 4342–4353 (2016) have been added to the revised manuscript. More explanations of FTIR, UV/Vis and STEM images discussion have been added as well. The method for structural determination is explained in Section 2.4. The BET surface area, pore volume and diameter were confirmed by N2 adsorption-desorption experiments. Lastly, McKay said that “For all of the systems studied, chemical reaction seems significant in the rate-controlling step and the pseudo-second order chemical reaction kinetics provides the best correlation of the experimental data.” [R1]. On the other hand, the nanocomposites were beneficial for selective adsorption of cationic dyes. Generally, energy absorption, electron transfer and electrostatic interactions play an intrinsic role in the process of luminescence, i.e., electrostatic and p···p interactions between the catalyst and dye molecules [R2]. Because of the polyoxometalate anion in the nanocomposites, we speculated that the adsorption of these compounds might due to the electrostatic interactions between adsorbents and cationic dye molecules.
[R1] Ho, Y.S.; McKay, G. Pseudo-second order model for sorption processes, Process Biochem. 1999, 34, 451–465.
[R2] Hou, Y.X.; Sun, J.S.; Zhang, D.P.; Qi, D.D.; Jiang, J.Z. Porphyrin-alkaline earth MOFs with the highest adsorption capacity for methylene blue. Chem. Eur J. 2016, 22, 6345–6352.

Reviewer 2 Report
The manuscript describes another interesting work on dye removal by Li and Liu. The manuscript has a significant amount of data, which is well-organized and well-described. The following comments could help to further improve the manuscript prior to publishing.
Justification for the selection of the components of the nanomaterials should be provided at the start of the manuscript. The manuscript has typos and grammatical mistakes that hinder conveying the scientific message. It is highly recommended that the manuscript is revised by a native speaker. What was the rationale for the feed concentrations? Are they practically relevant and represent dye concentrations in industrial wastewater, or contaminated ground water? Justification and a reference should be provided in the manuscript. The reproducibility of the materials fabrication should be demonstrated in the manuscript. Neither the characterization nor the application has shown any reproducibility results. Where possible error bars / standard deviations should be added. The figure captions are too short. Elaborate so that the figures caption are more informative, and the figures can be interpreted on their own without having to read the full text. The introduction should provide a general context for the work but currently it is narrowly limited to dyes. For instance, the authors should mention that there is a trend for sustainable wastewater treatment and give some examples (DOIs: 10.1021/acssuschemeng.9b02658; 10.1039/C5GC01937K; 10.1021/acsanm.9b00022; 10.1021/acsanm.8b01249). The supplier, purity and grade of all materials, chemicals and solvents should be listed under section 2.1 under the experimental part. Avoid using the ambiguous x/y format for the units, and replace them with the x y^-1 format, which is recommended by the IUPAC. The closely related research published by the authors should be acknowledged in the manuscript (DOI: 10.3390/nano8090710). The recycle of adsorbents is crucial requiring careful design of the elution/regeneration step (DOI: 10.1039/C6PY01853J), which should be mentioned by the authors. It is good that recycling with minor efficiency loss over 7 cycles were proved, and should be mentioned that further optimization of the elution could further improve the recycling. The y axis of Figure 9 seems incorrect. The maximum is 1%, or the authors meant 100%? Why is some text highlighted in red? It should be removed before resubmission. The conclusion section should give a quantitative summary of the main research findings.Author Response
Response: Thank you for your letter and comments concerning our manuscript. These comments are all valuable and very helpful for revising and improving our paper. Revised portion are marked in red in the paper. We have done our best to improve the quality of English and typing errors. This research was carried on the laboratories. So the practically relevant and represent dye concentrations are in distilled water. The reproducibility of the materials can be found in Section 3.5. Meantime, we went through some relevant literatures. Most of them didn’t show the bars/standard deviations. The figure captions have been supplemented. The introduction has provided a general context for the work. The related references (DOIs: 10.1021/acssuschemeng.9b02658; 10.1039/C5GC01937K; 10.1021/acsanm.9b00022; 10.1021/acsanm.8b01249; 10.3390/nano8090710; 10.1039/C6PY01853J) have been added to the revised manuscript. The supplier, purity and grade of all materials, chemicals and solvents have been listed in Section 2.1. All units have been changed to the x y^-1 format. It have been mentioned that further optimization of the elution could further improve the recycling. Figure 9 has been redrawn. The paper was submitted to Nanomaterials firstly. However, the editor suggested us to resubmit to Molecule after revising with highlighted. They have been removed. The conclusion section has given a quantitative summary of the main research findings.

Round 2
Reviewer 1 Report
The manuscript is ready for publication
Reviewer 2 Report
The authors have addressed the comments, and this reviewer recommends accepting the manuscript for publication after correcting the following small mistakes:
1. "curves shows" to "curves show"
2. "play an important role" to "play important roles"
3. Figure 8b unit should follow the format of the rest of the manuscript: x y^-1
4. "Kecilic" to "Kecili" in ref 48